# Comprehensive Land Consolidation as a Tool to Promote Rural Restructuring in China: Theoretical Framework and Case Study

Qiqi Yin [1,2], Shenglu Zhou [1,2,*], Chengxiang Lv [1,2], Yang Zhang [3], Xueyan Sui [3] and Xiaorui Wang [3]

1 School of Geographic and Ocean Science, Nanjing University, Nanjing 210023, China
2 Key Laboratory of Coastal Zone Exploitation and Protection, Ministry of Natural Resources, Nanjing 210024, China
3 Jiangsu Land Consolidation and Rehabilitation Center, Nanjing 210017, China
* Correspondence: zhousl@nju.edu.cn

**Abstract:** In the context of current global rural decline, land consolidation has been adopted with the objectives of promoting rural vitalization and regional sustainable development. In this paper, we provide a theoretical framework for rural restructuring driven by comprehensive land consolidation (CLC). The framework describes three key mechanisms of rural spatial, economic, and social restructuring driven by CLC: improving spatial patterns and functions, vitalizing the collective economy, and reshaping the social community. Based on the theoretical framework, we present a case that exemplifies the micro processes of rural restructuring. Taking spatial restructuring as the material basis and carrier, CLC promotes economic restructuring from traditional agricultural production to modern agricultural production and industrial integration, as well as social restructuring from a traditional rural society to urbanization, communitization, and a society with diversified culture. After CLC, it is very important to further enhance the sustainability of the collective economic development and enhance the cohesion and prosperity of the social community.

**Keywords:** comprehensive land consolidation; rural restructuring; theoretical framework; mechanisms; China

## 1. Introduction

With rapid urbanization, industrialization, and technological change, rural areas are experiencing a decline in agricultural production, population loss, and weakening rural vitality; as a result, rural decline has emerged as a global trend [1]. In response to this rural decline, many countries and regions require promoting the transformation and reconstruction of rural areas to break through the bottleneck of rural development. For example, Germany implements village renewal, diversification, and agritourism strategies to enhance the quality of life in rural areas and encourages the diversification of economic activity [2]. Japan carries out the "sixth industrialization" to promote the deep integration of agricultural resources and industries [3]. In China, the government proposed the Rural Vitalization Strategy to stimulate rural development based on five guidelines: prosperous life, industrial prosperity, ecological livability, rural civilization, and effective governance. In the process of policy implementation, rural areas are experiencing drastic restructuring and transformation: agricultural production is changing from extensive agriculture to market-intensive agriculture [4]; the productive value of rural territory has shifted to diversified values, such as a better environment and local cultural consumption; and rural land-use tends to be multifunctional [5].

Land consolidation is an excellent tool to implement rural development projects with multiple purposes and goals [5–8]. Relevant scholars believe that land consolidation is not "just" about creating conditions for efficient agricultural production, but also for providing conditions for rational labor distribution, urban–rural integration, regeneration of a land fund, improvement of resource management, and ecological stability [7–12]. Therefore,

land consolidation plays an important and unique role in rural transformation and rural restructuring. Since the 21st century, land consolidation has developed into an indispensable part of rural development strategy in countries with a long history of land consolidation, such as Germany and Poland [13]. In China, land consolidation is undergoing a transformation from traditional modes to comprehensive land consolidation (CLC). Although the theoretical and practical exploration of CLC in China has a ten-year history, the stage of relatively well-developed planning and implementation has only emerged in recent years. As marked by the <Notice on The Pilot Work of Comprehensive Land Consolidation in The Whole Region> issued in 2019, the government formally proposed to implement the national pilot project of CLC in no less than 300 villages and towns by 2020.

As a comprehensive measure, the multi-dimensional effects of CLC on rural space utilization [10,14–16], rural economic development [9,17–20], and social development [21,22] have attracted a lot of attention. These studies explored the internal processes and hidden mechanisms of CLC, promoting multi-dimensional rural development through empirical research at multiple regional scales. Other articles have discussed the theoretical system of CLC, promoting rural restructuring and rural revitalization, and established theoretical frameworks from different perspectives [23–25]. However, in view of problem orientation, existing research on the key mechanisms of rural restructuring driven by CLC in plain agricultural areas needs to be further strengthened. A plain agricultural area is an area where agriculture (planting) is the main production and cultivated land resources are abundant; it is the national grain and main agricultural products supply area. At present, plain agricultural areas are facing multiple pressures, such as agricultural economic recession, social organization degradation, farmland fragmentation, and ecological quality decline. Implementing CLC to promote the transformation and development of plain agricultural areas is an objective requirement and effective measure to achieve food security, human–land relationship coordination, and sustainable development in the new era.

The purpose of this study is to refine the key mechanisms of CLC to promote rural restructuring based on the problem orientation in plain agricultural areas; to build a theoretical framework; and, through a case study, conduct case empirical testing. The organizational structure of this study is as follows: First, based on exploring the connotation of rural restructuring and summarizing the main problems in rural development in China, this paper establishes a theoretical framework to explain the mechanisms of rural restructuring driven by CLC; then, a detailed description of a typical case is provided, outlining the micro process and sustainable pathways of rural restructuring driven by CLC, ending in a discussion and conclusion. Through theoretical analysis and case exemplification, we can provide a blueprint for the formulation of rural development policies.

## 2. A Theoretical Framework of Rural Restructuring Driven by CLC

### 2.1. Literature Review: The Connotation of Rural Restructuring

Rural restructuring explores the spatial and temporal changes in rural areas, involving multiple dimensions, such as the economy, society, regime, ecology, and culture [26,27], which is regarded as qualitative changes in the rural economic structure, social structures, and social practices. Rural change is the most extensive, including all changes that occur under the regulation of macro-economic policies, government expectations, and the natural evolution of civil society. Hoggart [28] emphasized that rural restructuring means something more than "change"; it involves an interactive process of multi-dimensional changes rather than a change in one "sector", where the processes of change are causally linked. Moreover, only when the changes accumulate beyond a numerical boundary can they be called restructuring; that is, to find a sense of "big change". Therefore, rural restructuring emphasizes two characteristics: the interactive and holistic transformation of multiple dimensions in rural areas, and a "big change" beyond a certain boundary.

So, where should rural restructuring go? Long and Tu [29] pointed out that the process of rural evolution refers to the slow progress, leap development, transient recession, rejuvenation, and other statuses in the time dimension of rural areas; that is, it not only refers to

the presence of positive progression, but also negative degradation. Rural restructuring refers to positive evolution: in order to adapt to changes in the kernel system and the external system of rural development, local actors integrate and reallocate key resources, such as population, land, and industry based on an evaluation of local conditions, so as to achieve the structural optimization and functional improvement of the rural territorial system [30,31]. Therefore, rural restructuring emphasizes effective human intervention, such as integrating economic factors, optimizing spatial structure, improving public facilities, etc. Rural restructuring is the process of realizing rural transformation [27]. Rural transformation could be considered as an analogous process, with the characteristics of final customs rather than an ongoing process [32]; therefore, rural transformation can be regarded as the cumulative outcome of rural restructuring. Under the paradigm of agricultural political economy, Western academia, especially British geographers, summarized the nonlinear restructuring process of rural areas from productivism to post-productivism; rural areas in developed Western countries have experienced a transformation from production oriented–consumption oriented–multifunctional–global rural [33–35].

From a systemic perspective, rural restructuring results from the joint drive of internal and external factors in a rural regional system. Expanding the time frame of the analysis reveals that urbanization, industrialization, informatization, and globalization have greatly affected the fundamental reconfigurations of various fields of rural life [36,37]. For example, technological development has affected rural areas by reducing employment in the primary sector, mainly through mechanization and increased productivity. The trend in industrialization has led to large-scale manufacturing entering rural areas, and the service sectors are shifting from cities to rural areas [38,39]. Oksa [40] sees key restructuring dimensions in Nordic rural areas relating to the penetration of information technology, the provision of public services, a new source of livelihood for farmers, and higher levels of personal mobility. These macro trends have led to two central results of rural restructuring: a decline in agriculture and urbanization. Nandi and Mistri [41] found that the current trend in rural transformation is from rural settlements to urban settlements in India, and the main driving forces are the expanding settlements, the increasing population density, and the improvement in public services. Hedlund and Lundholm [36] believe that the economic restructuring of the British countryside entailed a transformation from agricultural employment to manufacturing and then to the urban service sector. Urbanization not only promotes the modification of the rural economy but also has an important impact on landscape and community as once homogeneous and stable communities become dynamic and heterogeneous. Nelson [42] described rural restructuring in the western United States, believing that rural economy restructuring has led to new changes in human–land relations and increased local cultural diversity.

Further, since the early 1990s, the Western world has entered a new stage of capitalism, marked by post-Fordism in industry and neoliberalism in governance [43]. The amenity-led development has gained greater importance in rural areas, and the consumption of health care, leisure, and education has become the main driving force of rural restructuring and economic growth, prompting people to actively create landscapes and service sectors. Popular concerns about "the environment" are seen to be complementary, triggering the growth of new economic sectors and economic phenomena such as farm tourism, health care, and the second home [44–46], and fueling the process of rural population reflow and counter-urbanization [47]. This is illustrated by developments in agriculture itself [44] and the shift in people's focus from quantity to quality of the agricultural products (e.g., "slow" food and organic food), which have triggered the transformation and upgrading process of the agricultural industry.

Due to the dynamic interaction of endogenous and exogenous forces, the whole process of formation and operation of rural restructuring consists of plastic, multi-level interactions rather than isolated phenomena [48]. Zasada et al. [49] believe that rural assets, resources, and factor endowment are the key internal factors affecting rural restructuring, and the formulation of rural development policy should be an investment in regional

capital and capacity building, including physical, human, and natural capital, to adjust and diversify rural economic activities, promote the upgrading of traditional agriculture, and create additional income. If the external driving forces have engendered the general trend of rural transformation and restructuring towards consumption-oriented and multi-function economy, the formulation of rural development policies should actively adjust and restructure rural development elements while complying with the trend [48–50]. Several case studies [51–53] show that rural restructuring via CLC is a combination of "bottom-up" market-driven measures and "top-down" policy-driven measures. The government's policies play a role in regulating land use, public space governance, and infrastructure popularization, while local entrepreneurs, rural elites, and small businesses use these policies to vitalize the local economy.

### 2.2. The Theoretical Framework of Rural Restructuring Driven by CLC

CLC is driven by a series of macro-development strategies (such as rural vitalization, new urbanization, ecological construction, etc.) and derived from traditional land consolidation. With the accelerated development of industrialization, urbanization, and agricultural modernization, the traditional land consolidation model of a single factor, single measure, and single objective has been unable to solve complex problems such as disordered spatial layout, inefficient use of land resources, and degradation of ecological quality in rural areas. In this scenario, CLC has become an important platform for getting rid of rural development problems and breaking through bottlenecks. The implementation object of CLC is the whole rural area, based on regional background conditions, and is guided by the rural development strategy formulated in village planning, recommending diversified measures such as agricultural land consolidation, construction land consolidation, ecological restoration, and public space governance. Another diversified feature of CLC is the diversity of participants. The traditional model led by the government has gradually transformed into the model of consultation and co-construction by the government, villagers, enterprises, and local elites. The multiple stakeholders not only cooperate in the process of project implementation but also participate in the subsequent industrial development endeavor. Therefore, CLC pays attention to extending its industrial chain and value chain according to local conditions. Institutional guarantee is the backbone of CLC, and the implementation of planning, integration of factors, motivation, and balance of interests between multiple subjects are inseparable from the maintenance and support of both formal and informal institutions.

Long and Liu [26] point out that the restructuring of rural land use, rural industry, and rural social organization is pivotal to resolve rural developing problems, which constitute the three aspects of rural restructuring: spatial restructuring, economic restructuring, and social restructuring. Taking breaking the dilemma in rural development as the logical starting point, we explain the key mechanisms of how CLC promotes rural restructuring from three dimensions: space, economy, and society (Figure 1). Spatial restructuring is the material basis for and carrier of social and economic restructuring, and the object of CLC implementation is rural space. The goal of space restructuring is to improve the supply, bearing, and support functions of production–living–ecological spaces. Economic restructuring plays a leading role in rural development, including the flow and combination of production factors and the transformation and upgrading of industrial structures, as well as in improving the value of space utilization and providing vitality for social restructuring. Rural society is the subject of the construction and development of economic restructuring and spatial restructuring. Social restructuring is driven by spatial restructuring and economic restructuring; it is the embodiment of the change in residents' lifestyle, livelihood, and social relations.

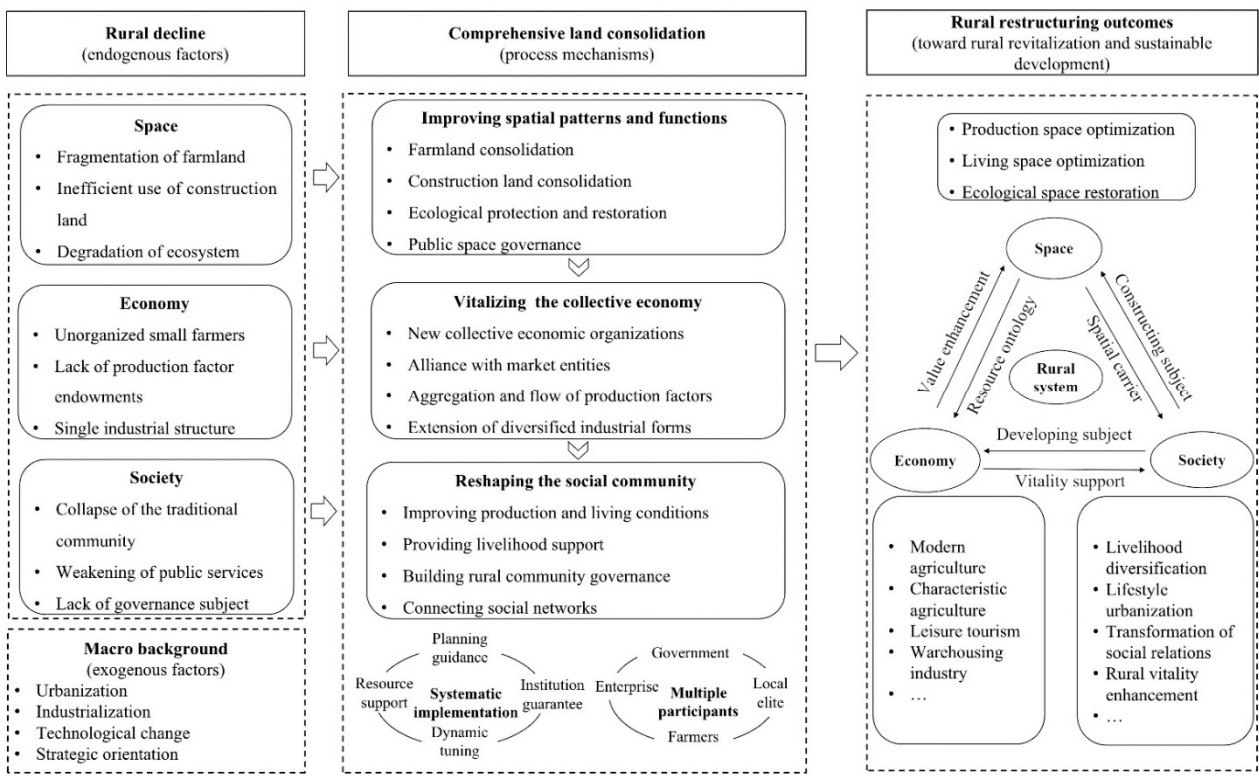

**Figure 1.** Theoretical framework of rural restructuring driven by comprehensive land consolidation.

### 2.2.1. Space Restructuring: Improving Spatial Patterns and Functions

Since the implementation of the household contract responsibility system in 1978, although the production model of distributing output to households has greatly improved the subjective enthusiasm of farmers, the government has conducted regular and irregular land redistribution to compensate for changing demographics, which, coupled with the restrictions of natural and geographical conditions, have resulted in the long-term trend of farmland fragmentation in China [54]. To a certain extent, it causes a waste of land resources, an increase in agricultural production costs, and a decrease in production efficiency. At the same time, the utilization of construction land also has the problems of inefficiency, low construction costs, moving costs, and land-use costs, leading to the inefficient use of homesteads by villagers. With the loss in labor, a bad evolution process of "external expansion and internal empty" of rural construction land and an increase in idleness and abandonment are created [55]. Despite the homestead's expansion, no improvements will have been made to its rural infrastructure and social services, and the lack of reasonable collective planning of the newly built villages damages the overall pattern and landscape of the countryside.

In view of the fragmentation of cultivated land, inefficient land use, degradation of ecological quality, and other problems, CLC optimizes the spatial layout in accordance with the principles of the relative concentration of space and optimal allocation of resources, accelerates the establishment of a spatial system for coordinating urban and rural areas, and enhances supply, carrying and supporting the functions of the production–living–ecological space. In terms of production space, the structure and layout of cultivated land are optimized, and the land productivity is enhanced through agricultural infrastructure construction and soil improvement. In terms of living space, the idle homestead is recovered and unified, so as to promote the efficient and intensive utilization of rural construction land to form a village–town pattern with a reasonable and orderly spatial layout and improved production and living environment and facilities. In terms of ecological space, the implementation of environmental protection and restoration helps to promote the regional ecological cycle and the construction of ecological networks, and

green infrastructure enhances the stability of the ecosystem and realizes the overflow of rural ecological functions and the complementarity of urban and rural functions.

### 2.2.2. Economic Restructuring: Vitalizing the Collective Economy

After solving the problem of space utilization, it is more important to help traditional rural areas break the "poverty trap equilibrium" and move towards a new "benign development equilibrium". In the past decades, under the urban–rural dual system, the drastic impact of modern industry, and an export-oriented economy, rural areas have evolved into a passive role of exporting cheap land and labor to the city. The capital accumulation of industrial and urban development comes at the cost of agricultural depletion and rural hollowing-out [56]. With the collapse of agricultural production cooperatives, the government withdrew from the direct control of and intervention in agricultural production, and the villagers turned from mutual aid unions to looser groups. The fragmentation of land and the lack of production knowledge, technology, and capital have increasingly highlighted the vulnerability of small-scale farmers' production, resulting in the atomization dilemma, whereby micro individuals occupy resources dispersedly and are unable to seize the opportunity to develop industries [57].

The key measure to adjust the economic structure of traditional agricultural villages is to regard the utilization of rural resources as a regional economy, and then develop new collective economic organizations by gathering production factors. The most important tasks of CLC is to rearrange rural land resources, so it is an appropriate moment for the organization of a collective economy. The government, enterprises, village collectives, and farmers participate in the process of rural land capitalization by means of competition or cooperation, which, although based on different utilization goals and benefit orientations, effectively promotes the integration and flow of capital, technology, labor, and other factors in rural areas. Different stakeholders jointly cultivate new collective economic organizations, such as cooperatives, agricultural enterprises, and large farms. These new collective economic organizations represent the formation of a community of interests. By tapping local advantageous agricultural resources and releasing industrial development space, the community further improves the industrial socialized service system, extends the agricultural industrial chain, adjusts the industrial structure, and develops new business forms, such as leisure agriculture, facility agriculture, health care, logistics, and warehousing [15,25,30,58]. Thus, the process of economic restructuring from traditional agricultural production to modern agricultural production and industrial integration has been realized.

### 2.2.3. Social Restructuring: Reshaping the Social Community

Since the reform and "opening up", the livelihood mode of small farmers in China has evolved from "work for agriculture" to "agriculture for industry". Farmers go to cities to work at leisure and return to the village during the busy season (in farming). Although they have engaged in "contractual" labor, they still retain their original social identity and wander between urban and rural areas [59]. The rural resident population generally presents the status of "aged, weak and child", rural collective organization is declining day by day, and governance is lacking, resulting in the loss of rural traditional culture, the fragmentation of social relations, a decline in public services, and a plethora of other social problems.

CLC aims to improve the production and living environment of rural residents by encouraging the transformation of living spaces from the original single and mixed form to a more diversified, centralized, and three-dimensional development, thus further promoting the process of social restructuring in terms of residents' quality of life, livelihood, social networks, and other aspects of their lives [60]. Through the establishment of new communities, the rebuilding of grass-roots organizations, and the improvement of grass-roots governance, local urbanization of rural inhabitants is encouraged—rural inhabitants move to counties and towns near their native villages and their way of life and production have been fundamentally transformed by urbanization [61]. The residents' livelihood capital has

changed from simple agriculture to a diversified and non-agricultural state, and residents are typically spontaneous, pluralistic, and professional in their work. A rural society that had tended to be stable, closed, and rigid is gradually broken and replaced by a more fluid, open, and complex "multi-pattern". The social relations among residents, which are mostly based on geographical and blood relations, evolve into a compound relationship of geography, blood, industry, organization, and market relations. On the basis of mutual interest, the consensus of mainstream values, and the support of cooperation mechanisms, rural inhabitants in the new community gradually form a new social community.

## 3. Typical Case Study: Gaozuo Town

### 3.1. Study Area and Data Collection

#### 3.1.1. Study Area

Gaozuo Town is located in the northwest of Yancheng City, Jiangsu Province, China. It belongs to Yangtze River Delta plain agricultural area and covers an area of 6933 ha, of which 4267 ha are cultivated land, accounting for 61.2% of total land area. Traditional planting agriculture is the leading industry of the town. The crops mainly consist of rice and wheat, and are supplemented by lobster, turkey and other animal breeding, lotus root, water bamboo, and other economic crops. The town is 9.5 km away from the center of the county and its territory is rich in water resources. A small number of secondary and tertiary industries have been developed. Land used for commercial services comprises less than 1% of the total area, and industrial land 7.7%. The secondary industries are mostly low-end industries, including mechanical processing, brick and tile factories, poultry processing plants, and additional tertiary industries that need to be developed.

Gaozuo Town is facing the problems of labor loss and population aging. From 2010 to 2020, the permanent population decreased by around 22%. In 2020, there were 18,959 permanent residents, including 12,178 over the age of 50, accounting for 64.23% of the total population. In total, 52% of the villagers are part-time workers, mainly working in factories in the town or nearby counties, or self-employed. Gaozuo Town has been involved in CLC projects since 2018 and was selected in 2019 as one of the 20 national comprehensive land improvement pilots in Jiangsu Province. From 2018 to 2020, Gaozuo Town has successively promoted rural housing relocation, agricultural land consolidation, land transfer, and other work. By 2020, nine villages represented by Jidun village have completed CLC.

#### 3.1.2. Data Collection

The data collection in this paper mostly consists of unstructured interviews conducted to obtain first-hand information, as well as a large number of internal documents and public reports, such as the implementation plan of CLC, village planning, as well as local population, economic, and social statistical data. Unstructured interviews are a powerful qualitative research method to gather valid information and develop new insights into research topics [62]. The research group visited Gaozuo Town in January 2022 and March 2022 to conduct centralized research. First, open-ended interviews were conducted with leaders and elites of the relevant departments of the town government (including town secretary, town mayor, project organization members, and cadres of the Agricultural Bureau) and local large agricultural enterprises, to obtain data and material on the progress of project implementation, rural industrial development planning, and the operation process, among other things. Then, in-person interviews were conducted with residents in two typical new rural communities (Guangming Family Community and Happy Home Community), to obtain data and material on the productivity and living conditions of residents. After that, we kept WeChat contact with the project organization personnel and the interviewed villagers, and further obtained research data through follow-up visits.

### 3.2. Processes of Rural Restructuring Driven by CLC

3.2.1. Spatial Restructuring

Production space: Compared with the fragmented agricultural land before consolidation, the agricultural land now tends to be large-scale and centralized. Before consolidation, the total area of farmland was 4083.8 ha, with an average household area of 0.67 ha. Farmland infrastructure does not adapt to modern agricultural production. After consolidation, the farmland increased by 507.1 ha, mainly from the demolition and reclamation of homesteads (215.9 ha), reclamation of old ditches between plots (115.2 ha) and rural roads (50.3 ha), and the filling of abandoned ponds (125.1 ha) (Table 1). In order to realize the large-scale operation of agriculture, the land ownership of farmers is checked uniformly, and the land transfer is organized, so that the scattered land management rights are centralized in the village collective. In order to improve land productivity, a series of well-equipped farmland projects have been carried out, including high-standard irrigation and drainage planning, soil fertility improvement, field road, and farmland shelterbelt projects. The non-agricultural industrial land has not changed, but a cold-chain logistics center, agricultural science and technology research, and additional equipment have been introduced to the agricultural product-processing industrial park. The public service facilities in the tourist area were upgraded, and improvements include the construction of ecological parking lots and visitor centers to enhance the service level of the tourist area.

**Table 1.** Land-use transfer matrix in the case study area (unit: ha).

| Before Consolidation | After Consolidation | | | | | | | |
|---|---|---|---|---|---|---|---|---|
| | Farmland | Facility Agricultural Land | Non-Agricultural Industrial Land | Traffic Land | Residential Land | Forest and Grass Land | Water Area | Sum |
| Production space | | | | | | | | |
| Farmland | 4083.8 | | | | | | | 4083.8 |
| Facility agricultural land | 115.2 | 188.7 | | | | | | 303.9 |
| Non-agricultural Industrial land | | | 74.8 | | | | | 74.8 |
| Traffic land | 50.3 | | | 123.7 | | | | 174.0 |
| Living space | | | | | | | | |
| Residential land | 215.9 | | | | 96.3 | | | 312.1 |
| Ecological space | | | | | | | | |
| Forest and grass land | 0.7 | | | | | 532.8 | | 533.5 |
| Water area | 125.1 | | | | | | 866.5 | 991.6 |
| Sum | 4590.9 | 188.7 | 74.8 | 123.7 | 96.3 | 532.8 | 866.5 | 6473.7 |

Living space: The living space has been reduced by 215.9 ha after consolidation (Table 1), and the per capita construction land area was reduced by 75%. The living space of most farmers has been reorganized after consolidation. Before consolidation, the homestead was mainly close to rivers and roads, which were distributed in a disorderly "一" or "非" shape, mixed with production and ecological space (Figure 2). In total, 75% of the homesteads were built in the 1980s; the architectural form was generally old, and the overall style was "small, scattered, empty and broken". After consolidation, the rural homesteads were removed, merged, and relocated, and the living space concentrated (Figure 2). The new communities are located near the county seat, and the houses are apartments and villas, complete with water, electricity, gas, and other facilities. The community plans the public living space of residents in a unified way, supporting public infrastructure such as a fitness square, a leisure park, and a day care center, and the sanitation and fire-fighting infrastructure is nearly equivalent to that of cities. Families that have not been relocated usually have no changes in housing, transportation, sewage treatment system, or other facilities.

Ecological space: Due to the expansion of cultivated land area, the area of ecological space has been reduced by 125.8 ha after consolidation. The reduced ecological space mostly consists of water bodies (Table 1), but the ecological function of the space has been significantly improved. Before consolidation, some river sections had a lot of garbage, black and smelly sediment, and the self-purification function of the water body was damaged. Ecological space restoration focused on the two main rivers; the river section with serious

siltation shall be dredged to improve the drainage capacity. After the implementation of the project, the drainage standard will be increased from once in 10 years to once in 10–20 years. These biological treatment measures have been implemented to improve water quality, build a composite ecosystem through an ecological floating bed and bottom material improvement, and form healthy and perfect ecological wetland spots in local rivers and ditches, which will improve the long-term absorption of water pollution and restore the water body's self-purification capacity.

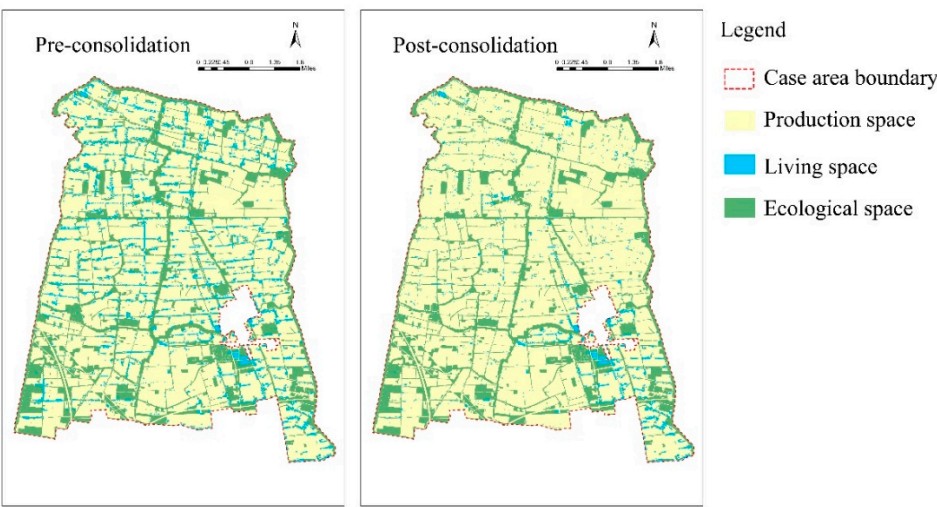

**Figure 2.** Spatial utilization before and after comprehensive land consolidation in Gaozuo town.

3.2.2. Economic Restructuring

During the implementation of CLC, the town first tapped its potential internally, selecting grassroots cadres and rural elites with high education, agricultural production knowledge, and technology knowhow, and who were familiar with industrial management, to form the backbone of rural economic development. To centralize land management, land-share cooperatives were established in each village. When high willingness and demand can be confirmed, villagers were assisted with carrying out a land transfer at a 10–15% higher value than comparable properties in the surrounding areas. For the land of farmers who join the cooperative, the land shares were calculated and paid according to the land area actually owned by farmers, but the specific plot and location of land are not determined. This method effectively improved the enthusiasm of villagers for land transfer.

As a new type of collective economic organization, cooperatives distributed and implemented the agricultural production. In this process, collective economic organizations only own land and lack production factors such as capital, technology, and management. In order to achieve modern agricultural development, cooperatives cooperated with the town's agricultural company and transferred the land management rights to the company, creating a jointly planned, modern agricultural production system. The town's agricultural company hired farmers back to work in the agricultural park, and provided capital, technology, management, and other means of production. Under the leadership of the town's agricultural company, the cooperatives worked together to provide unified management, procurement, sales, advertising, and income distribution of agricultural production. Villagers and cooperatives distributed the income of the modern agricultural park according to the number of land shares, and the residual income belongs to the company. The cooperative pays secondary dividends to the villagers according to land equity. The joint win–win model of "companies + cooperatives + farmers" has gradually taken shape (Figure 3).

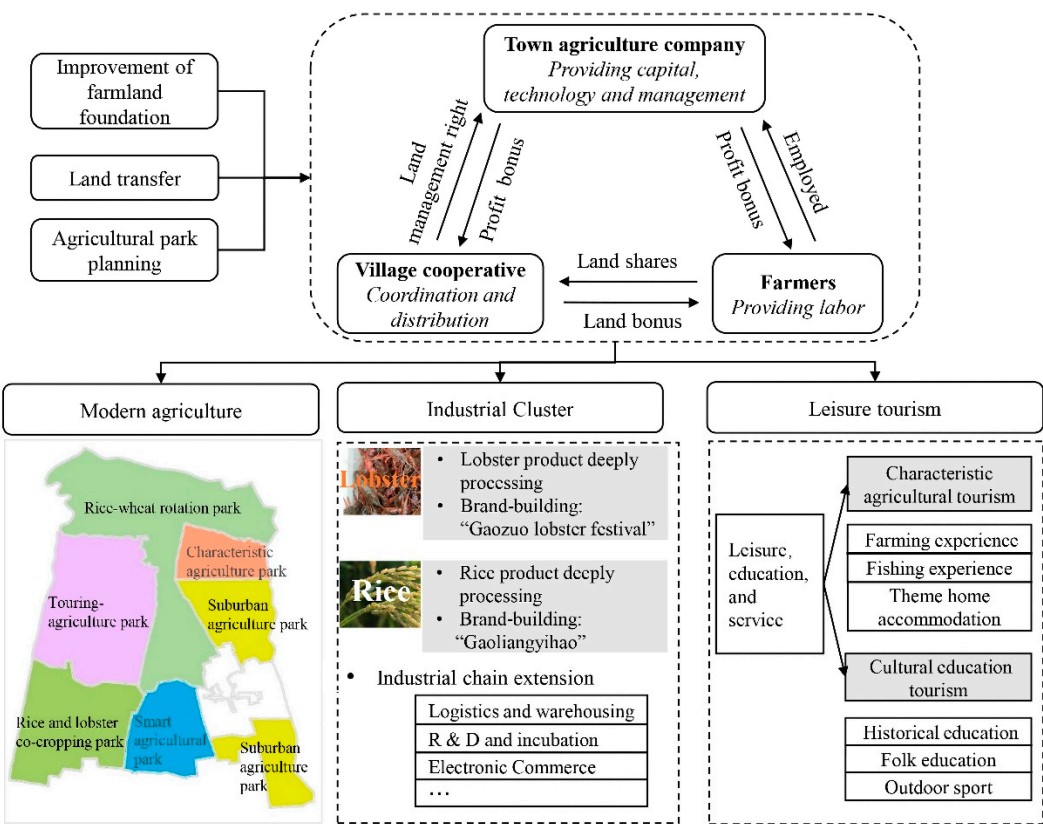

**Figure 3.** Mode of multi-stakeholder alliance and integrated development of industries in Gaozuo town.

Tapping into its agricultural foundations, the town has adopted lobster and high-quality rice as its key industries, and relies on the village's history, culture, and fisherman's landscape to form an industrial integration development mode of "modern agriculture + industrial cluster + leisure tourism." The agricultural production of the town is divided into six production parks (Figure 3). Relying on the primary products provided by the park, processing industries for lobster, rice, turkey, and other characteristic agricultural products have been developed to enhance the added value of the agricultural products and extend the application to other industries such as logistics warehousing, product incubation, and e-commerce operations. Building on the local history, culture, and traditional agricultural resources, the town has fulfilled the cultural and tourism needs of the county, expanded rural leisure tourism industries, such as agricultural culture education and fishery breeding experience, and provided historical and outdoor sports locations for residents. The development of diversified industries has increased the collective commercial income of Jidun village by 220% in 2020 compared with 2017.

### 3.2.3. Social Restructuring

Changes in quality of life: After consolidation, 50% of the residents chose to move to the new rural settlement built by the government, 40.6% of the residents chose cash demolition compensation to buy a house in the city, and 9.3% of the residents chose to remain at their original address. The new settlement is equipped with a comprehensive service center, fitness activity square, elderly care service center, health room, and other amenities (Figure 4a–d). The public space had been afforested and upgraded for roads, and public service facilities such as environmental sanitation facilities, parking lots, and legal bulletin boards have been set up (Figure 4e–g). Before consolidation, villagers usually owned courtyards to plant self-sufficient grains and vegetables. In order to help residents adapt to the new living environment, the community left a small area for each family to grow vegetables (Figure 4h). With the change in living environment, residents' daily

lifestyles have become more urbanized, and they are satisfied with the living environment of the community. Centralized residences had not changed the rural Hukou system. They have retained the rural family registration and all collective economic rights. At the same time, they also have a social security system different from the cities, including pension, life insurance, education, and medical treatment. Respondents interviewed noted:

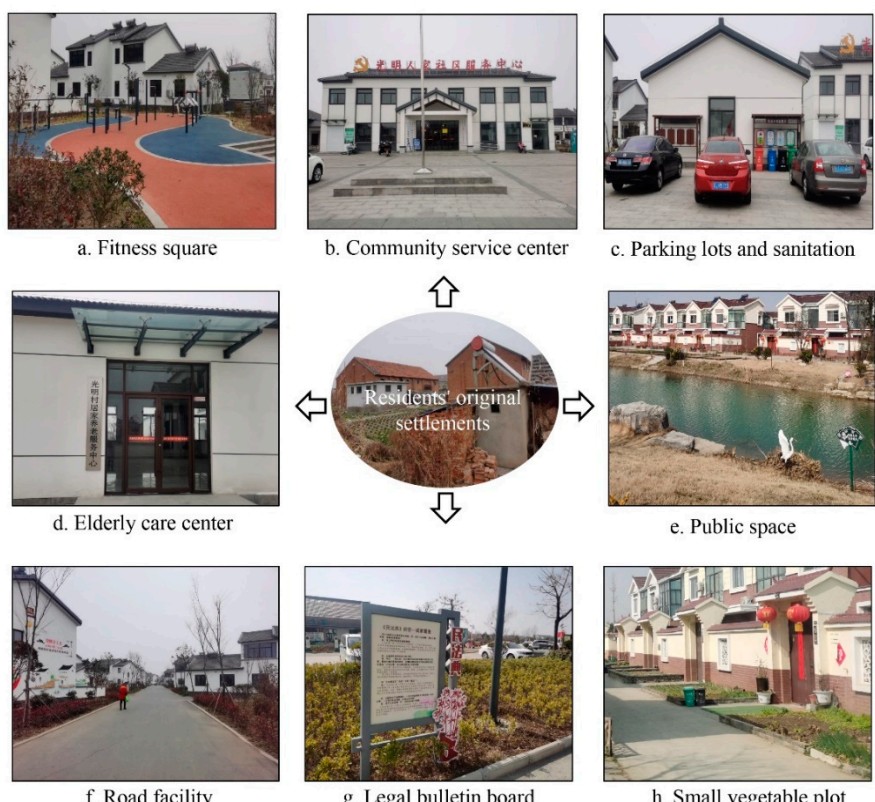

**Figure 4.** Living facilities and public services in the new settlement. Pictures (**a**–**h**) show the living facilities and public services in two typical new rural communities (Guangming Family Community and Happy Home Community); the middle picture shows the living facilities in the original rural settlements. Source: The first author took the photographs in 2022.

> *"The greening and housing in the community have been significantly improved compared with the original ones. Now the living conditions are the same as those in the city. There are places for shopping, entertainment and business."*

> *"I now get nearly 2000 yuan of pension every year. If I get sick, the village will send 500 yuan of consolation money. There is also a cooperative medical security."*

Transformation of livelihood mode: Before consolidation, the main income sources of rural inhabitants were farming and migrant work, and the income structure was relatively simple. After consolidation, the income structure gradually became more diversified and came from sources such as land share dividends, working income in the agricultural zone, individual income from tourism, and migrant work. In 2020, the per capita disposable income of inhabitants in Jidun village was 21,181 yuan, an increase of 50.3% over its income before consolidation. The agricultural zones have prioritized employing local inhabitants for production, and the farmers have been transformed from smallholders to employed agricultural producers.

> *"Our 0.53 ha land is transferred to the collective at the price of 12,750 yuan per ha. I usually look after my grandchildren at home and work in the zone in my spare time. Compared with the grain planting, I can earn 25,000 yuan more every year." "In the past, the income from grain planting was low and the labor intensity was high. Now the zone uses mechanized farming uniformly, which is easier than before."*

The new settlement was built closer to the center of the county, so the inhabitants who have moved into the community have tended to go into the county to engage in non-agricultural work. However, some residents believed that moving into the new community was not good for them, and that the loss of self-sufficient land affected the security of their family livelihood. One expressed his idea by saying that:

*"My wife and I have been working in a factory near the village. We could rely on our own land to grow grain and vegetables and raise some fowls to achieve self-sufficiency. After relocation, we have to buy all from the market, and our daily consumption also increases. The demolition compensation could only solve the problem in the short term. In the long run, our livelihood is even more difficult."*

Changes in social relations: The integrated removal and resettlement way enabled the inhabitants to retain their original social relations, but also add new social relations. The community has provided activities such as free movies, public canteens, and book houses to enrich the daily lives of the resettled inhabitants, and the social network has expanded.

*"I often practice dancing with the sisters of the art team and get to make more friends. When I first came here, I was not used to living in the community. Now I really regard this place as my home."*

At the same time, the relationship between community management organizations is gradually strengthening. Before consolidation, most of the inhabitants left the village to work throughout the year and participated little in village affairs. After consolidation, the community encouraged the inhabitants to participate in daily management activities such as road cleaning and public security, and set up a consultation system to jointly solve community management problems with the inhabitants.

The farmers who have not moved into the new community are mainly the elderly, who were highly attached to their hometown.

*"We have been living here for a long time. Now we are more than 60 years old and do not want to live away from home. We are not used to living the houses in the community."*

For young people, their hometown was the place of their childhood memories; they have supported the relocation efforts, but also have their own views on the relocation.

*"The old trees and ancient wells in the village were the source of happiness in my childhood. Now there is no such scene in the new community. We suggest that some special buildings be properly preserved when the old village is demolished, so that we can have a place to remember our hometown."*

## 4. Discussion

Much as is the case with the rural recession experienced in Europe and North America, China's rural areas are undergoing a complex process of urbanization, rural population reduction, and industrialization [63–65]. Underpinning the new era, the CLC, which promotes rural economic and social restructuring on the basis of spatial restructuring, is strongly systematic and goal-orientated, and is highly consistent with the requirements of national social and economic development and transformation, making it an important opportunity for the realization of rural leapfrog development.

As a new form of land consolidation under the background of the new era, the novelty of CLC lies in the construction of a regional sustainable economy on the basis of spatial consolidation projects. A collective economy has great potential in organizing the reuse of abandoned land, strengthening the exchange of knowledge and other elements, providing more employment opportunities, and enhancing collective marketing income, and has been advocated in many developed and developing countries [66–68]. Several studies have used empirical case analysis combining qualitative and quantitative methods and found that the collective economic model of the village cooperative and social enterprise unified management promoted the extension of the agricultural industry chain and local economic

growth; additionally, the residents' livelihood structure became non-agricultural and diversified [17,18,61,69,70]. However, a diachronic test is required to determine whether the new collective economy can form a sustainable development model. Studies [53,71] suggested that the sustainable development of industries in some areas where CLC has been implemented is still uncertain due to the lack of competition and limited sources of capital. Local governments must formulate strategies to continuously attract enterprises, give appropriate pricing rights to industry owners and managers, and follow market changes and consumer preferences [1]. Due to the lack of local production factors in rural areas, the development of a collective economy needs the assistance of external "others", such as large capital enterprises, to provide the necessary financial and technical support. Although the adoption of a market-oriented agricultural production model contributes to the growth of the overall economy, it may not take into account the food security and livelihood of families/individuals. Research on Africa [72–74] found that land-use consolidation negatively impacted the food security of small farmers and aggravated socio-economic differentiation due to autocratic implementation and single planting of specific crops. The manipulation of land resources by the market will inevitably have a great impact on indigenous people, who are regarded as "us". Therefore, it is necessary to not only make the development of the collective economy meet the needs of marketization, urbanization, and internationalization, but also guard against the phenomenon of "others" seizing space, prevent the loss of rights and interests of farmers, and ensure that the fruits of the collective economy are shared by "us".

In order for a region to achieve long-term prosperity, it is very important to establish villagers' recognition of their new local residential system. Several studies have focused on the impact of the implementation of CLC on interpersonal relationships and identity. They found that the social network of the residents expanded in the new settlements, most resettled villagers' social relations were place-bound, and their existing relationships with neighbors and friends were well maintained after land consolidation [22,75]. Land consolidation inspired farmers' enthusiasm to participate in rural government, and their identity cognition gradually changed from one of bystanders to decision-makers and supervisors [53,76]. In contrast, some communities were faced with the breakdown of the residents' original social network and lifestyle changes that have posed a great challenge to their social lives, particularly for the older generation [77]. After the transition to the new community, traditional social relations established by rural geography and kinship gradually disintegrate, and changes in livelihood, space, and people contribute to a longstanding state of uncertainty about the future for the local inhabitants. A study in northern Brazil showed that although conglomerate farming through land consolidation has improved the productivity and profits of large farms, the poverty of local small farmers, landless workers, women, and indigenous people has worsened [78]. The vulnerability of farmers' livelihoods was associated with insecure land rights, reduced access to common shared resources, fewer on-farm job opportunities, the loss of self-sufficiency, and obstacles to finding new employment opportunities [78–82]. In addition to providing good housing, public facilities, pensions, and other social security for landless farmers, it is important for the government to provide reasonable agricultural and non-agricultural employment opportunities, and create an environment where they are encouraged to use their own abilities, realize their potential, and eventually flourish.

## 5. Conclusions

In the context of the current global rural decline, land consolidation is needed to promote rural vitalization and regional sustainable development. This study puts forward a theoretical framework of CLC-driven rural spatial, social, and economic restructuring, and then describes the micro-processes through a case study. The spatial pattern of the case area has changed from a mixed and interwoven production, living, and ecological space, to an orderly cluster of intensive utilization, and the spatial function has been enhanced. The economic form has changed from a typical "dual economy" of migrant

workers and traditional planting and breeding to a "diversified economy" that integrates the development of three types of industries. The social structure has changed from a traditional rural society with geographical and blood ties to a society with urbanization, community, culture, and diversified governance, with geographical, blood, industry, and market ties.

CLC is a systematic, regional, and multidimensional measure that optimizes the rural system through government intervention. It is expected to be used as both a tool and a platform, as well as a component of the Chinese government's broader mandate to bring about the vitalization of traditional agricultural villages. We believe that China's CLC plays an increasingly important role in promoting rural development and has achieved some positive results, which can provide reference and enlightenment for other countries and regions in the world that plan to promote rural renewal and rural vitalization. Further research is necessary, for example, on how to strengthen the long-term development and sustainability of the rural collective economy and inhabitants' livelihood after the implementation of CLC.

**Author Contributions:** Conceptualization, Q.Y. and S.Z.; data curation, Q.Y. and C.L.; funding acquisition, S.Z. and Y.Z.; investigation, Q.Y., C.L., X.S. and X.W.; methodology, Q.Y.; project administration, S.Z., Y.Z. and X.W.; software, C.L.; writing—original draft, Q.Y.; writing—review and editing, Q.Y., S.Z., Y.Z. and X.S. All authors have read and agreed to the published version of the manuscript.

**Funding:** This research was supported by the National Natural Science Foundation of China (41771243) and the Natural Resources Science and Technology Project of Jiangsu Province, China (2020001, 2020006).

**Institutional Review Board Statement:** Not applicable.

**Informed Consent Statement:** Not applicable.

**Data Availability Statement:** Not applicable.

**Conflicts of Interest:** The authors declare no conflict of interest.

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
