# Peer review of "Comprehensive Land Consolidation as a Tool to Promote Rural Restructuring in China: Theoretical Framework and Case Study"

_land, doi:10.3390/land11111932_

Round 1

Reviewer 1 Report

In general, this paper explored an essential topic regarding comprehensive land consolidation as a tool to promote rural restructuring. The topic falls into the scope of Land very well. However, it has some problems in the manuscript.

(1)  This literature review lacks in-depth research. This manuscript just focuses on the connotation of rural restructuring; it is too simple. The situation and driving mechanism of rural restructuring should also be reviewed because it is relevant to your research topic. Also, the paper needs to be better embedded in the international literature.

(2)  The theoretical framework should be improved. Why can the rural system be classified into these three aspects? Furthermore, the content has not clarified the relationship between the three aspects. Moreover, some grammar errors exist in Figure 1, and the quality of figures (almost every figure) presented in the manuscript is below average.

(3) The discussion just covers the results in this version. The authors should compare the manuscript to existing research in terms of research method, research perspective, conclusions, and so on.

(4) There are some grammar errors in the figures and contents, which implies that the authors need another round of careful English editing.

Author Response

Dear Editors and Reviewers:

Thank you for your letter and for the reviewers’ comments concerning our manuscript. Those comments are all valuable and very helpful for revising and improving our paper, as well as the important guiding significance to us researches. We have studied comments carefully and have made correction which we hope to meet with your approval. Revised portions are marked in red in the paper. The main corrections in the paper and the responds to the reviewer’s comments are as flowing:

Reviewer 1#

In general, this paper explored an essential topic regarding comprehensive land consolidation as a tool to promote rural restructuring. The topic falls into the scope of Land very well. However, it has some problems in the manuscript.

R: Thanks for your valuable comments. Based on your valuable comments, we have carefully revised the manuscript. We addressed and answered all suggested and requested points. Revised portions are marked in red in the manuscript. In addition, the manuscript has been edited for structure, logic, flow, language, grammar, and improved clarity by using the Journal English Language Editing service. Some long original sentences were cut down to better transfer the conception to the readers.

(1)  This literature review lacks in-depth research. This manuscript just focuses on the connotation of rural restructuring; it is too simple. The situation and driving mechanism of rural restructuring should also be reviewed because it is relevant to your research topic. Also, the paper needs to be better embedded in the international literature.

R: Thanks for your valuable comments. We have revised as you suggested. (P3-4, Line 119-168)

We added the literature review of the situation and driving mechanism of rural restructuring. The main content is to divide the driving mechanism of rural restructuring into external factors (urbanization, industrialization, informatization and globalization) and internal factors (rural resources endowment and policy implementation).

In addition, we have enriched the international literature studies in the literature review, and some simple and duplicate contents were cut down and revised.

From a systemic perspective, rural restructuring results from the joint drive of internal and external factors in the rural regional system. Expanding the time frame of the analysis reveals that urbanization, industrialization, informatization, and globalization have greatly affected the fundamental reconfigurations of various fields of rural life [36, 37]. For example, technological development has affected rural areas by reducing employment in the primary sector, mainly through mechanization and increased productivity. The trend of industrialization has led to large-scale manufacturing entering rural areas, and the service sectors are shifting from cities to rural areas [38, 39]. Oksa [40] sees key restructuring dimensions in Nordic rural areas relating to the penetration of information technology, the provision of public services, the new source of livelihood for farmers and higher levels of personal mobility. These macro trends have led to two central results of rural restructuring: the decline of agriculture and urbanization. Nandi and Mistri [41] found that the current trend of rural transformation is from rural settlements to urban settlements in India, and the main driving forces are the expanding settlements, the increasing population density, and the improvement of public services. Hedlund and Lundholm [36] believe that the economic restructuring of the British countryside entailed a transformation from agricultural employment to manufacturing and then to the urban service sector. Urbanization not only promotes the modification of the rural economy but also has an important impact on landscape and community as once homogeneous and stable communities become dynamic and heterogeneous. Nelson [42] described rural restructuring in the western United States, believing that rural economy restructuring has led to new changes in human-land relations and increased local cultural diversity.

Rural restructuring emphasizes the systematic change of multiple dimensions. Oksa [36] sees key restructuring dimensions in Nordic rural areas relating to the penetration of information technology, the provision of public services, the new source of livelihood for farmers and higher levels of personal mobility. Fuller and van den Bor [37] argued that rural restructuring is a synchronized process can be divided into four aspects: geopolitical change, economic restructuring, social change and environmentalism. The economic restructuring is the main storyline of rural restructuring, including the decline of agriculture and related sectors and the rise of new departments. Hedlund [38] believes that the economic restructuring of British countryside had experienced the transformation from agricultural employment to manufacturing, and then to urban service sector; Tonts [39] argued that the trajectory of rural economic restructuring can be characterized by the employment and unemployment transfer in different departments. In addition to economic restructuring, other related key elements in restructuring include significant changes in rural culture, family well-being, social relationships, and so on. Tigges [40] argued that the restructuring of places leads to and is defined by the restructuring of social relationships, including family and marital relations, neighboring and class relationships. Nelson [41] described rural restructuring in the western United States from three dimensions: altered human-land relationships, in and out migration, shift in economic sectors, and paying attention to the interaction between economic restructuring and the increasing cultural diversity in rural areas. In China, there are many studies focusing on rural spatial restructuring [15, 42], especially rural settlement restructuring, which are mainly related to the implementation of CLC [43, 44]. More attention is paid to the relationship between spatial restructuring and socio-economic restructuring, such as the change of residents' social relations [45], and the impact of spatial restructuring on rural labor mobility [46].

Further, since the early 1990s, the Western world has entered a new stage of capitalism marked by post-Fordism in the industry and neoliberalism in governance [43]. The amenity-led development has gained greater importance in rural areas, and the consumption of health care, leisure, and education has become the main driving force of rural restructuring and economic growth, prompting people to actively create landscapes and service sectors. Popular concerns about “the environment” are seen to be complementary, triggering the growth of new economic sectors and economic phenomena such as farm tourism, health care, and the second home [44-46] and fueling the process of rural population reflow and counter-urbanization [47]. This is illustrated by developments in agriculture itself [44] and the shift in people’s focus from quantity to quality of agricultural products (e.g., “slow” food and organic food), which have triggered the transformation and upgrading process of the agricultural industry.

Due to the dynamic interaction of endogenous and exogenous forces, the whole process of formation and operation of rural restructuring consists of plastic, multi-level interactions rather than isolated phenomena [48]. Zasada et al. [49] believe that rural assets, resources, and factor endowment are the key internal factors affecting rural restructuring, and the formulation of rural development policy should be an investment in regional capital and capacity building, including physical, human, and natural capital, to adjust and diversify rural economic activities, promote the upgrading of traditional agriculture and create additional income. If the external driving forces have engendered the general trend of rural transformation and restructuring towards consumption-oriented and multi-function economy, the formulation of rural development policies should actively adjust and restructure rural development elements while complying with the trend [48-50]. Several case studies [51-53] show that rural restructuring via CLC is a combination of "bottom-up" market-driven measures and "top-down" policy-driven measures. The government's policies play a role in regulating land use, public space governance, and infrastructure popularization, while local entrepreneurs, rural elites, and small businesses use these policies to vitalize the local economy.

(2)  The theoretical framework should be improved. Why can the rural system be classified into these three aspects? Furthermore, the content has not clarified the relationship between the three aspects. Moreover, some grammar errors exist in Figure 1, and the quality of figures (almost every figure) presented in the manuscript is below average.

R: Thanks for your valuable comments. We have revised as you suggested.

We expained the relationship between the three aspects: space, economy and society (P4-5, Line 191-206).

We improving the English language editing for Figure 1. Moreover, the English editing of all figures (Figure 1-4) were improved.

Long and Liu [26] point out that the restructuring of rural land use, rural industry, and rural social organization is pivotal to resolve rural developing problems, which constitute the three aspects of rural restructuring, i.e., spatial restructuring, economic restructuring and social restructuring. Take breaking the dilemma in rural development as the logical starting points, we explain the key mechanisms of how CLC promotes rural restructuring from three dimensions: space, economy and society (Fig.1). Spatial restructuring is the material basis for and carrier of social and economic restructuring, and the object of CLC implementation is rural space. The goal of space restructuring is to improve the supply, bearing, and support functions of production-living-ecological spaces. Economic restructuring plays a leading role in rural development, including the flow and combination of production factors and the transformation and upgrading of industrial structure, as well as in improving the value of space utilization and providing vitality for social restructuring. Rural society is the subject of the construction and development of economic restructuring and spatial restructuring. Social restructuring is driven by spatial restructuring and economic restructuring; it is the embodiment of the change in residents' lifestyle, livelihood, and social relations.

(3) The discussion just covers the results in this version. The authors should compare the manuscript to existing research in terms of research method, research perspective, conclusions, and so on.

R: Thanks for your valuable comments. We have revised as you suggested. We added the comparative analysis with existing international research (P13-14, Line 524-568).

Several studies have used empirical case analysis combining qualitative and quantitative methods and found that the collective economic model of the village cooperative and social enterprise unified management promoted the extension of the agricultural industry chain and local economic growth; additionally, the residents' livelihood structure became non-agricultural and diversified [17, 18, 61, 69, 70]. However, a diachronic test is required to determine whether the new collective economy can form a sustainable development model. Studies [53, 71] suggested that the sustainable development of industries in some areas where CLC has been implemented is still uncertain due to the lack of competition and limited sources of capital. Local governments must formulate strategies to continuously attract enterprises, give appropriate pricing rights to industry owners and managers, and follow market changes and consumer preferences [1]. Due to the lack of local production factors in rural areas, the development of collective economy needs the assistance of external "others" such as large capital enterprises to provide necessary financial and technical support. Although the adoption of a market-oriented agricultural production model contributes to the growth of the overall economy, it may not take into account the food security and livelihood of families/individuals. Research on Africa [72-74] found that land use consolidation negatively impacted the food security of small farmers and aggravated socio-economic differentiation due to autocratic implementation and single planting of specific crops. The manipulation of land resources by the market will inevitably have a great impact on indigenous people, who are regarded as "us". Therefore, it is necessary to not only make the development of collective economy meet the needs of marketization, urbanization and internationalization, but also guard against the phenomenon of "others" seizing space, prevent the loss of rights and interests of farmers and ensure that the fruits of collective economy are shared by "us".

In order for a region to achieve long-term prosperity, it is very important to establish villagers' recognition of their new local residential system. Several studies have focused on the impact of the implementation of CLC on interpersonal relationships and identity. They found that the social network of the residents expanded in the new settlements, most resettled villagers' social relations were place-bound, and their existing relationships with neighbors and friends were well maintained after land consolidation [22, 75]. Land consolidation inspired farmers' enthusiasm to participate in rural government, and their identity cognition gradually changed from one of bystanders to decision-makers and supervisors [53, 76]. In contrast, some communities were faced with the breakdown of the residents' original social network and lifestyle changes that have posed a great challenge to their social lives, particularly for the older generation [77]. After the transition to the new community, traditional social relations established by rural geography and kinship gradually disintegrate, and changes in livelihood, space, and people contribute to a longstanding state of uncertainty about the future for the local inhabitants. A study in northern Brazil showed that although conglomerate farming through land consolidation has improved the productivity and profits of large farms, the poverty of local small farmers, landless workers, women, and indigenous people has worsened [78]. The vulnerability of farmers' livelihoods was associated with insecure land rights, reduced access to common shared resources, fewer on-farm job opportunities, the loss of self-sufficiency, and obstacles to finding new employment opportunities [78-82].

(4) There are some grammar errors in the figures and contents, which implies that the authors need another round of careful English editing.

R: Thanks for your valuable comments. We have revised as you suggested. The manuscript has been edited for structure, logic, flow, language, grammar, and clarity by using the Journal English Language Editing service..We have shown the main revised sentences, and other parts of the article have also been edited. 

P27-29

With rapid urbanization, industrialization, and technological change, rural areas are experiencing a decline in agricultural production, population loss, and weakening rural vitality; as a result, rural decline has emerged as a global trend.

P224-228

CLC optimizes spatial layout in accordance with the principles of relative concentration of space and optimal allocation of resources, accelerates the establishment of a spatial system for coordinating urban and rural areas, and enhances supply, carrying and support functions of production-living-ecological space.

P252-254

The key measure to adjust the economic structure of traditional agricultural villages is to regard the utilization of rural resources as a regional economy, and then develop new collective economic organizations by gathering production factors.

P272-273

Farmers go to cities to work at leisure and return to the village at busy season (in farming).

P289-291

The residents' livelihood capital have changed from simple agriculture to a diversified and non-agricultural state, and residents are typically spontaneous, pluralistic, and professional in their work.

Special thanks for your good comments.

Reviewer 2 Report

Dear Authors,

The title of the study  “Comprehensive land consolidation as a tool to promote rural restructuring: theoretical framework and case study” corresponds to its content, but I suggest changing to:   “Comprehensive land consolidation as a tool to promote rural restructuring: theoretical framework and case study - China”.

Keywords:  comprehensive land consolidation; rural restructuring; theoretical framework; mechanisms; China are correct.

The total value of work is a valuable contribution. References take 64 publications are cited in the entire article. Literature research well started, but not enough publications. It is proposed to add the following articles that contain new research in this area, for example:

·         Basista, I.; Balawejder, M. 2020. Assessment of selected land consolidation in south-eastern Poland. Land Use Policy 2020, 99, 105033. https://doi.org/10.1016/j.landusepol.2020.105033

·         Stręk, Ż.; Noga, K. 2019. Method of delimiting the spatial structure of villages for the purposes of land consolidation and exchange. Remote Sens. 2019, 11, 1268. https://doi.org/10.3390/rs11111268

Similarly, the discussion or conclusion should refer to research conducted in this field in other countries and cited in this publication. Please complete this and the article will be a valuable scientific contribution.

Figure 2. Spatial utilization before and after CLC of the case area - not readable (blurry). Please improve the quality of figure 2.

For figure 1, 2, 3, 4 please correct the names. They are currently illegible.

It should be noted that the whole of the study is cognitive and contains important scientific elements. The article was written at a good academic level. In relation to the above, I express the opinion that the work submitted for review should be published in its entirety after taking into account the comments of the reviewer but not require a review again.

Author Response

Dear Reviewer:

Thanks for your valuable comments.Based on your valuable comments, we have carefully revised the manuscript. We addressed and answered all suggested and requested points. Revised portions are marked in red in the manuscript. In addition, the manuscript has been edited for structure, logic, flow, language, grammar, and improved clarity by using the Journal English Language Editing service. Some long original sentences were cut down to better transfer the conception to the readers. The main corrections in the paper and the responds to the reviewer’s comments are as flowing:

 (1) The title of the study “Comprehensive land consolidation as a tool to promote rural restructuring: theoretical framework and case study” corresponds to its content, but I suggest changing to:   “Comprehensive land consolidation as a tool to promote rural restructuring: theoretical framework and case study - China”. Keywords:  comprehensive land consolidation; rural restructuring; theoretical framework; mechanisms; China are correct.

R: Thanks for your valuable comments. We have revised as you suggested.

We revised the title of the study: “Comprehensive land consolidation as a tool to promote rural restructuring in China: theoretical framework and case study”.

 (2) The total value of work is a valuable contribution. References take 64 publications are cited in the entire article. Literature research well started, but not enough publications. It is proposed to add the following articles that contain new research in this area, for example:

  • Basista, I.; Balawejder, M. 2020. Assessment of selected land consolidation in south-eastern Poland. Land Use Policy 2020, 99, 105033. https://doi.org/10.1016/j.landusepol.2020.105033
  • Stręk, Ż.; Noga, K. 2019. Method of delimiting the spatial structure of villages for the purposes of land consolidation and exchange. Remote Sens. 2019, 11, 1268. 

R: Thanks for your valuable comments. We have revised as you suggested. We have increased the number of publications in the literature research, the literatures are mainly added in the “2.1 Literature review” and “4. Discussion”, and the number of references for the paper has increased from 64 to 82. We have enriched the content of the literature review and added international literature studies (P3-4, Line 119-168).

From a systemic perspective, rural restructuring results from the joint drive of internal and external factors in the rural regional system. Expanding the time frame of the analysis reveals that urbanization, industrialization, informatization, and globalization have greatly affected the fundamental reconfigurations of various fields of rural life [36, 37]. For example, technological development has affected rural areas by reducing employment in the primary sector, mainly through mechanization and increased productivity. The trend of industrialization has led to large-scale manufacturing entering rural areas, and the service sectors are shifting from cities to rural areas [38, 39]. Oksa [40] sees key restructuring dimensions in Nordic rural areas relating to the penetration of information technology, the provision of public services, the new source of livelihood for farmers and higher levels of personal mobility. These macro trends have led to two central results of rural restructuring: the decline of agriculture and urbanization. Nandi and Mistri [41] found that the current trend of rural transformation is from rural settlements to urban settlements in India, and the main driving forces are the expanding settlements, the increasing population density, and the improvement of public services. Hedlund and Lundholm [36] believe that the economic restructuring of the British countryside entailed a transformation from agricultural employment to manufacturing and then to the urban service sector. Urbanization not only promotes the modification of the rural economy but also has an important impact on landscape and community as once homogeneous and stable communities become dynamic and heterogeneous. Nelson [42] described rural restructuring in the western United States, believing that rural economy restructuring has led to new changes in human-land relations and increased local cultural diversity.

Further, since the early 1990s, the Western world has entered a new stage of capitalism marked by post-Fordism in the industry and neoliberalism in governance [43]. The amenity-led development has gained greater importance in rural areas, and the consumption of health care, leisure, and education has become the main driving force of rural restructuring and economic growth, prompting people to actively create landscapes and service sectors. Popular concerns about “the environment” are seen to be complementary, triggering the growth of new economic sectors and economic phenomena such as farm tourism, health care, and the second home [44-46] and fueling the process of rural population reflow and counter-urbanization [47]. This is illustrated by developments in agriculture itself [44] and the shift in people’s focus from quantity to quality of agricultural products (e.g., “slow” food and organic food), which have triggered the transformation and upgrading process of the agricultural industry.

Due to the dynamic interaction of endogenous and exogenous forces, the whole process of formation and operation of rural restructuring consists of plastic, multi-level interactions rather than isolated phenomena [48]. Zasada et al. [49] believe that rural assets, resources, and factor endowment are the key internal factors affecting rural restructuring, and the formulation of rural development policy should be an investment in regional capital and capacity building, including physical, human, and natural capital, to adjust and diversify rural economic activities, promote the upgrading of traditional agriculture and create additional income. If the external driving forces have engendered the general trend of rural transformation and restructuring towards consumption-oriented and multi-function economy, the formulation of rural development policies should actively adjust and restructure rural development elements while complying with the trend [48-50]. Several case studies [51-53] show that rural restructuring via CLC is a combination of "bottom-up" market-driven measures and "top-down" policy-driven measures. The government's policies play a role in regulating land use, public space governance, and infrastructure popularization, while local entrepreneurs, rural elites, and small businesses use these policies to vitalize the local economy.

We quoted the literatures you recommended in the appropriate chapter - “1. Introduction” (P2,59-60).

As a comprehensive measure, the multi-dimensional effects of CLC on rural space utilization [10, 14-16], rural economic development [9, 17-20], and social develpoment [21-22], has attracted a lot of attention.

15.Basista, I. and Balawejder, M., (2020). Assessment of selected land consolidation in south-eastern Poland. Land Use Policy, 99, 105033.

16. Stręk, Ż. and Noga, K., (2019). Method of Delimiting the Spatial Structure of Villages for the Purposes of Land Consolidation and Exchange. Remote Sensing, 11(11), 1268.

 (3) Similarly, the discussion or conclusion should refer to research conducted in this field in other countries and cited in this publication. Please complete this and the article will be a valuable scientific contribution.

R: Thanks for your valuable comments. We have revised as you suggested. We added the comparative analysis with existing international research in the discussion, and some duplicate contents were cut down and revised. (P13-14, Line 524-568).

Several studies have used empirical case analysis combining qualitative and quantitative methods and found that the collective economic model of the village cooperative and social enterprise unified management promoted the extension of the agricultural industry chain and local economic growth; additionally, the residents' livelihood structure became non-agricultural and diversified [17, 18, 61, 69, 70]. However, a diachronic test is required to determine whether the new collective economy can form a sustainable development model. Studies [53, 71] suggested that the sustainable development of industries in some areas where CLC has been implemented is still uncertain due to the lack of competition and limited sources of capital. Local governments must formulate strategies to continuously attract enterprises, give appropriate pricing rights to industry owners and managers, and follow market changes and consumer preferences [1]. Due to the lack of local production factors in rural areas, the development of collective economy needs the assistance of external "others" such as large capital enterprises to provide necessary financial and technical support. Although the adoption of a market-oriented agricultural production model contributes to the growth of the overall economy, it may not take into account the food security and livelihood of families/individuals. Research on Africa [72-74] found that land use consolidation negatively impacted the food security of small farmers and aggravated socio-economic differentiation due to autocratic implementation and single planting of specific crops. The manipulation of land resources by the market will inevitably have a great impact on indigenous people, who are regarded as "us". Therefore, it is necessary to not only make the development of collective economy meet the needs of marketization, urbanization and internationalization, but also guard against the phenomenon of "others" seizing space, prevent the loss of rights and interests of farmers and ensure that the fruits of collective economy are shared by "us".

In order for a region to achieve long-term prosperity, it is very important to establish villagers' recognition of their new local residential system. Several studies have focused on the impact of the implementation of CLC on interpersonal relationships and identity. They found that the social network of the residents expanded in the new settlements, most resettled villagers' social relations were place-bound, and their existing relationships with neighbors and friends were well maintained after land consolidation [22, 75]. Land consolidation inspired farmers' enthusiasm to participate in rural government, and their identity cognition gradually changed from one of bystanders to decision-makers and supervisors [53, 76]. In contrast, some communities were faced with the breakdown of the residents' original social network and lifestyle changes that have posed a great challenge to their social lives, particularly for the older generation [77]. After the transition to the new community, traditional social relations established by rural geography and kinship gradually disintegrate, and changes in livelihood, space, and people contribute to a longstanding state of uncertainty about the future for the local inhabitants. A study in northern Brazil showed that although conglomerate farming through land consolidation has improved the productivity and profits of large farms, the poverty of local small farmers, landless workers, women, and indigenous people has worsened [78]. The vulnerability of farmers' livelihoods was associated with insecure land rights, reduced access to common shared resources, fewer on-farm job opportunities, the loss of self-sufficiency, and obstacles to finding new employment opportunities [78-82].

 (4) Figure 2. Spatial utilization before and after CLC of the case area - not readable (blurry). Please improve the quality of figure 2.

R: Thanks for your valuable comments. We have revised as you suggested. We have improved the quality of figure 2 and redrawn it.

(5) For figure 1, 2, 3, 4 please correct the names. They are currently illegible.

R: Thanks for your valuable comments. We have revised as you suggested.

Figure 1. Theoretical framework of rural restructuring driven by comprehensive land consolidation.

Figure 2. Spatial utilization before and after comprehensive land consolidation in Gaozuo town.

Figure 3. Mode of multi-stakeholder alliance and integrated development of industries in Gaozuo town. 

Figure 4. Living facilities and public services in the new settlement.

It should be noted that the whole of the study is cognitive and contains important scientific elements. The article was written at a good academic level. In relation to the above, I express the opinion that the work submitted for review should be published in its entirety after taking into account the comments of the reviewer but not require a review again.

Thanks for your recognition of our research.

Special thanks for your good comments.

Round 2

Reviewer 1 Report

The research has been revised to a large extent and it's in a high quality. I believe that the authors put a lot of efforts into their research. I propose to accept the article in its present form.